

# Genome-wide identification of the bHLH transcription factor family in *Rosa persica* and response to low-temperature stress

Yueying Zhuang[1], Lijun Zhou[1], Lifang Geng[1], Lv Jiang[1], Yunji Sui[2], Le Luo[1], Huitang Pan[1], Qixiang Zhang[1] and Chao Yu[1]

[1] Beijing Key Laboratory of Ornamental Plants Germplasm Innovation & Molecular Breeding, National Engineering Research Center for Floriculture, Beijing Laboratory of Urban and Rural Ecological Environment, Key Laboratory of Genetics and Breeding in Forest Trees and Ornamental Plants of Ministry of Education, School of Landscape Architecture, Beijing Forestry University, Beijing, China
[2] Xinjiang Career Technical College, Xinjiang, China

Corresponding author
Chao Yu, yuchao@bjfu.edu.cn

## ABSTRACT

**Background:** Basic helix-loop-helix (*bHLH*) transcription factors are involved in plant growth and development, secondary metabolism, and abiotic stress responses have been studied in a variety of plants. Despite their importance in plant biology, the roles and expression patterns of bHLH family genes in *Rosa persica* have not been determined.

**Methods:** In this study, the *RbebHLH* family genes were systematically analyzed using bioinformatics methods, and their expression patterns under low-temperature stress were analyzed by transcriptome and related physiological index measurements.

**Results:** In total, 142 *RbebHLHs* were identified in the genome of *R. persica*, distributed on seven chromosomes. Phylogenetic analysis including orthologous genes in *Arabidopsis* divided *RbebHLHs* into 21 subfamilies, with similar structures and motifs within a subfamily. A collinearity analysis revealed seven tandem duplications and 118 segmental duplications in *R. persica* and 127, 150, 151, 172, and 164 segmental duplications between *R. persica* and *Arabidopsis thaliana*, *Prunus mume*, *Fragaria vesca*, *Rosa chinensis*, and *Prunus persica*, respectively. A number of cis-regulatory elements associated with abiotic stress response and hormone response were identified in *RbebHLHs*, and 21 *RbebHLHs* have potential interactions with the CBF family. In addition, the expression results showed that part of bHLH may regulate the tolerance of *R. persica* to low-temperature stress through the jasmonic acid and pathway. Transcriptomic data showed that the expression levels of different *RbebHLHs* varied during overwintering, and the expression of some *RbebHLHs* was significantly correlated with relative conductivity and MDA content, implying that *RbebHLHs* play important regulatory roles in *R. persica* response to low-temperature stress. Overall, this study provides valuable insights into the study of *RbebHLHs* associated with low-temperature stress.

## INTRODUCTION

The basic helix-loop-helix (*bHLH*) transcription factor family is the second largest transcription factor family in plants after the MYB transcription factor family. *bHLH* transcription factors play important roles in many physiological processes, such as plant growth, development, and secondary metabolism as well as the regulation of the maturation and senescence of carpel edge, pollen shell tissue, leaves, petals, and roots (*Hu et al., 2020*; *Zhu & Zhou, 2021*). They regulate the biosynthesis of flavonoids, such as anthocyanins, thus affecting flower and leaf color (*Park et al., 2007*; *Spelt et al., 2000*). In addition, *bHLH* transcription factors can regulate plant responses to abiotic stresses, including low temperatures, drought, salt stress, and iron deficiency stress (*Pei, 2017*; *Qian et al., 2021*).

The conserved *bHLH* domain contains about 60 amino acids and is composed of two parts. The basic region at the amino terminus is involved in DNA binding, and the helix-loop-helix region distributed at the carboxyl terminus promotes the interaction between proteins and forms homodimers or heterodimers (*Murre, McCaw & Baltimore, 1989*). *bHLH* transcription factors have been identified in many plant genomes, including 162 in the model plant *Arabidopsis thaliana* (*Heim et al., 2003*), 165 in rice (*Oryza sativa* japonica) (*Li et al., 2006*), 175 in apple (*Malus domestica*) (*Yang et al., 2017*), 226 in wild strawberry (*Fragaria vesca*) (*Folta et al., 2011*), and 66 in sweet cherry (*Prunus avium*) (*Shen et al., 2021*). Among ornamental plants, the plum (*P. mume*) and peach (*P. persica*) genomes each have 95 (*Zhang et al., 2018*; *Ding et al., 2021*).

*Rosa persica*, also known as *Rosa berberifolia* (*Zhang et al., 2021a*), is a perennial deciduous shrub belonging to the subgenus Hulthemia in the family Rosaceae. It is a second-class protected plant in China, where it is only distributed in some areas north of the Tianshan Mountain in Xinjiang. It mostly grows in shrubland communities in arid areas, such as slopes, wastelands, or roadsides at altitudes of 120–950 m in the form of constructed species (*Zhu, 2003*). China is the distribution center of *Rosa* and has abundant resources.

Owing to the limited distribution of *R. persica* and other factors, current research on *R. persica* is mainly focused on the potential suitable area, biological characteristics (*Hui, Zhang & Wang, 2013*), microstructure (*Hui, Zhang & Wang, 2014*), and other individual-or community-level properties. Owing to its good biological characteristics, such as cold resistance, drought resistance, and saline-alkali tolerance, as well as other key traits, such as single leaves without stipules, yellow flowers with purple and red floral patches, and well-developed roots, *R. persica* is a valuable gene resource for resistance breeding in *Rosa*. Only the effect of external calcium on salt stress has been explored in studies of the resistance of *R. persica* (*Yan et al., 2022*), and the excellent resistance and variegated traits of *R. persica* have not been deeply explored at the molecular level. In this study, bioinformatics methods were used to identify *bHLHs* in the whole genome of *R. persica* and to analyze their phylogenetic relationships, physical and chemical properties, gene localization, and collinearity. Based on the analysis of the cis-acting elements in the promoter region of *bHLHs*, 45 genes with more cis-acting elements were

selected and their expression levels during the overwintering period were analyzed. In addition, the potential regulatory target genes of some *bHLHs* were predicted. This study lays the foundation for further analysis of potential mechanisms of *bHLHs* in response to low-temperature stress in *R. persica* and screening of excellent candidate genes for cold resistance.

# MATERIALS AND METHODS

## Identification of members of the *RbebHLH* family

The genome sequence data, gene structure annotation information, and functional information for *R. persica* were obtained from our previous research (unpublished). The AtbHLH protein sequence was downloaded from the plant transcription factor database PlantTFDB (http://planttfdb.gao-lab.org). Using the TBtools Blast Wrapper tool (*Chen et al., 2020*), based on the bHLH gene family reference sequence of *A. thaliana*, possible sequences in *R. persica* were retrieved with an E-value threshold of $<1 \times 10^{-5}$. NCBI Batch CD Search (https://www.ncbi.nlm.nih.gov/Structure/bwrpsb/bwrpsb.cgi) (*Marchler-Bauer et al., 2015*) was used for searches based on the conserved *bHLH* gene family structure domain. Detection of conserved structural domains in sequences by HMMER (https://www.ebi.ac.uk/Tools/hmmer/) and Pfam databases (http://pfam.xfam.org/). Genes that did not contain *bHLH* domains were excluded. TBtools Fasta Stats was used to evaluate the CDS length, the length of the protein, and chromosomal location information. Expasy PI (https://web.expasy.org/compute_) online tools were used to evaluate molecular weight (MW) and protein isoelectric (pI) points. The online tool WoLF PSORT (https://wolfpsort.hgc.jp) (*Horton et al., 2007*) was used for subcellular localization prediction. TBtools was used to visualize the chromosomal location and tandem repeat genes of *RbebHLHs*.

## Multiple sequence alignment and gene structure analysis

MEGA (https://www.megasoftware.net/) and Jalview (https://www.jalview.org/) were used to generate a bHLH protein sequence alignment, evaluate known conserved sites, and explore new conserved sites. MEME was used to analyze the conserved motifs (*Bailey & Elkan, 1994*). The motif length was set to 6–200 amino acid residues, the maximum number of motifs was 10, and default values were used for other parameters. The NCBI CD-Search (https://www.ncbi.nlm.nih.gov/Structure/cdd/wrpsb.cgi) online tools to predict the conserved motifs for functional annotation. Structural information was obtained using the *R. persica* genome annotation file, and the results were visualized using TBtools to draw the distribution of conserved motifs and a gene structure map.

## Phylogenetic and gene duplication analysis

*AtbHLH* transcription factors from the *A. thaliana* TAIR database (https://www.arabidopsis.org) (*Reiser et al., 2017*) were downloaded, including data for proteins encoded by 162 At*bHLHs*. The Muscle function in MEGA was used to generate a multiple sequence alignment of bHLH proteins of *Arabidopsis* and *R. persica*. 'JTT+G' was identified as the optimal amino acid substitution model for the phylogenetic analysis (*Nei & Kumar, 2000*).

The FastTree function (*Price, Dehal & Arkin, 2009*) in Geneious Prime was used to construct the maximum likelihood evolutionary tree (1,000 bootstrap replicates), and iTOL (https://itol.embl.de) (*Letunic & Bork, 2021*) was used to display the phylogenetic tree. The chromosomal location of each gene was obtained from the GFF file of the *R. persica* genome. Additionally, genome data for *A. thaliana*, *P. mume*, *P. persica*, *F. vesca*, and *R. chinensis* were downloaded from the NCBI database. BLASTp (*Mahram & Herbordt, 2015*) and MCScanX (*Wang et al., 2012*) were used to analyze gene duplication events. Additionally, TBtools was used to calculate the non-synonymous substitution rate ($K_a$), synonymous substitution rate ($K_s$), and their ratio for homologous bHLH pairs in *R. persica* and *R. chinensis* to evaluate the type of selection pressure acting on protein-coding genes. Based on λ (*i.e.*, the substitution rate per synonymous site per year), $K_s$ values were converted to divergence times (T) in millions of years. Circos and collinearity diagrams were drawn using TBtools to visualize the gene density and the collinearity.

## Cis-acting element analysis and transcriptional target gene network prediction

The GFF Sequence Extractor tool in TBtools was used to extract the 2,000 bp sequence upstream of the *bHLH* transcription factor start codon from the *R. persica* annotation file. PlantCare (http://bioinformatics.psb.ugent.be/webtools/plantcare/html/) online tools were used to predict *RbebHLH* regulatory elements in gene promoter regions. The HeatMap in TBtools was used to draw a statistical map of the distribution of homeopathic elements. The JASPER database (https://jaspar.genereg.net/) was used to obtain transcription factor binding motif information, and MEME FIMO was used to predict target genes. Cytoscape was used to map the network of interactions between transcription factors and target genes.

## Plant material and qRT-PCR analysis

Monocotyledonous rose material was harvested from Hutubi County, Xinjiang, China, next to National Highway 312. From mid-September to mid-April, the main roots (about 10 cm from the root base) and mature stems (about 15 cm from the stem base) were collected from Hutubi under natural overwintering conditions at intervals of 1 month, and more than 30 g of each part was sampled each time. The collected material was fixed in liquid nitrogen for 30 s, and then transported back to the laboratory on dry ice to maintain a low-temperature cold chain for storage in an ultra-low temperature refrigerator. Samples were used for RNA extraction, complementary DNA (cDNA) library construction, and transcriptome sequencing based on the *R. persica* reference genome. Raw data obtained by sequencing have not yet been published. The expression level was calculated and normalized according to the fragments per kilobase per million mapped reads (FPKM). The TBtools cartoon heat map was used to visualize the gene expression levels.

RNA was extracted using an RNA Extraction Kit (Omega Bio-tek, Norcross, GA, USA). cDNA was then synthesized using the PrimeScript RT Reagent Kit with gDNA Eraser (Takara, Kusatsu, Japan). Quantitative real-time PCR(qRT-PCR) experiments were

performed using the TB Green premix Ex Taq II Kit using the qTOWER2.2 system (Analytik Jena, Jena, Germany). Three biological and three technical replicates were performed for each sample. All primers are shown in Schedule 5. The reaction system consisted of 25 µL of TB Green Premix Ex Taq II (12.5 µL), forward and reverse primer mix (2 µL), cDNA sample (2 µL), and sterilized water (8.5 µL). The reaction program was as follows: initial denaturation at 95 °C for 30 s, followed by 40 repeated cycles at 95 °C for 5 s, 55 °C for 30 s, and 72 °C for 30 s. Relative gene expression levels were calculated using the $2^{-\Delta\Delta Ct}$ method (*Zhang et al., 2021b*) using the *RbeGAPDH* (glyceraldehyde-3-phosphate dehydrogenase) gene as an internal reference.

## Physiological index measurement method

The relative conductivity was determined as follows: Wash the test material with tap water and absorb it, cut it into small pieces about 1 cm long, mix it well, put 1 g of plant in each test tube, and add 10 mL of deionized water. Place the tubes in a vacuum desiccator, pump for 10–15 min, deflate and shake lightly for 20 min, and measure the conductivity of the extracts with a conductivity meter. The conductivity of the deionized water blank was also measured. The conductivity was measured by placing the measured tubes in a boiling water bath at 100 °C for 15 min and then cooling down for 10 min before measuring the conductivity after boiling. Relative conductivity = (sample conductivity-blank conductivity)/(boiling conductivity-blank conductivity) × 100%. The malondialdehyde content was determined using the thiobarbituric acid colorimetric method (*Hodges et al., 1999*). Five biological replicates and three technical replicates were performed. Correlation analysis of physiological indicators and expression of selected genes was performed using Origin 2021.

## RESULTS

### Identification of members of the *RbebHLH* family

By a Blast search and verification against the NCBI CD database, 337 candidate *bHLH* sequences in the *R. persica* genome were obtained. Genes that did not contain the conserved bHLH domain were eliminated, resulting in 142 *RbebHLHs*, named *RbebHLH1–RbebHLH142* (Table S1). Of these, 140 *RbebHLHs* were distributed on seven chromosomes. Two *RbebHLHs* failed to be assembled to the chromosomal level. Proteins were between 73 (*RbebHLH33*) and 960 (*RbebHLH13*) amino acids (aa), with a predicted MW of between 7643.82 (*RbebHLH33*) and 104532.56 (*RbebHLH13*). The predicted PI was between 4.42 (*RbebHLH134*) and 11.79 (*RbebHLH111*). The subcellular localization predicted by WoLF PSORT showed that 117 *bHLHs* were located in the nucleus, 12 in the chloroplast, six in the cytoplasm, three in the cytoplasm and nucleus, and one in the mitochondria, peroxisome, Golgi, and endoplasmic reticulum.

### Multiple sequence alignment and gene structure analysis of *RbebHLHs*

The *bHLH* region is located at the C-terminus of the *bHLH* domain, which is mainly composed of a hydrophilic lipid helix composed of hydrophobic residues and a loop and

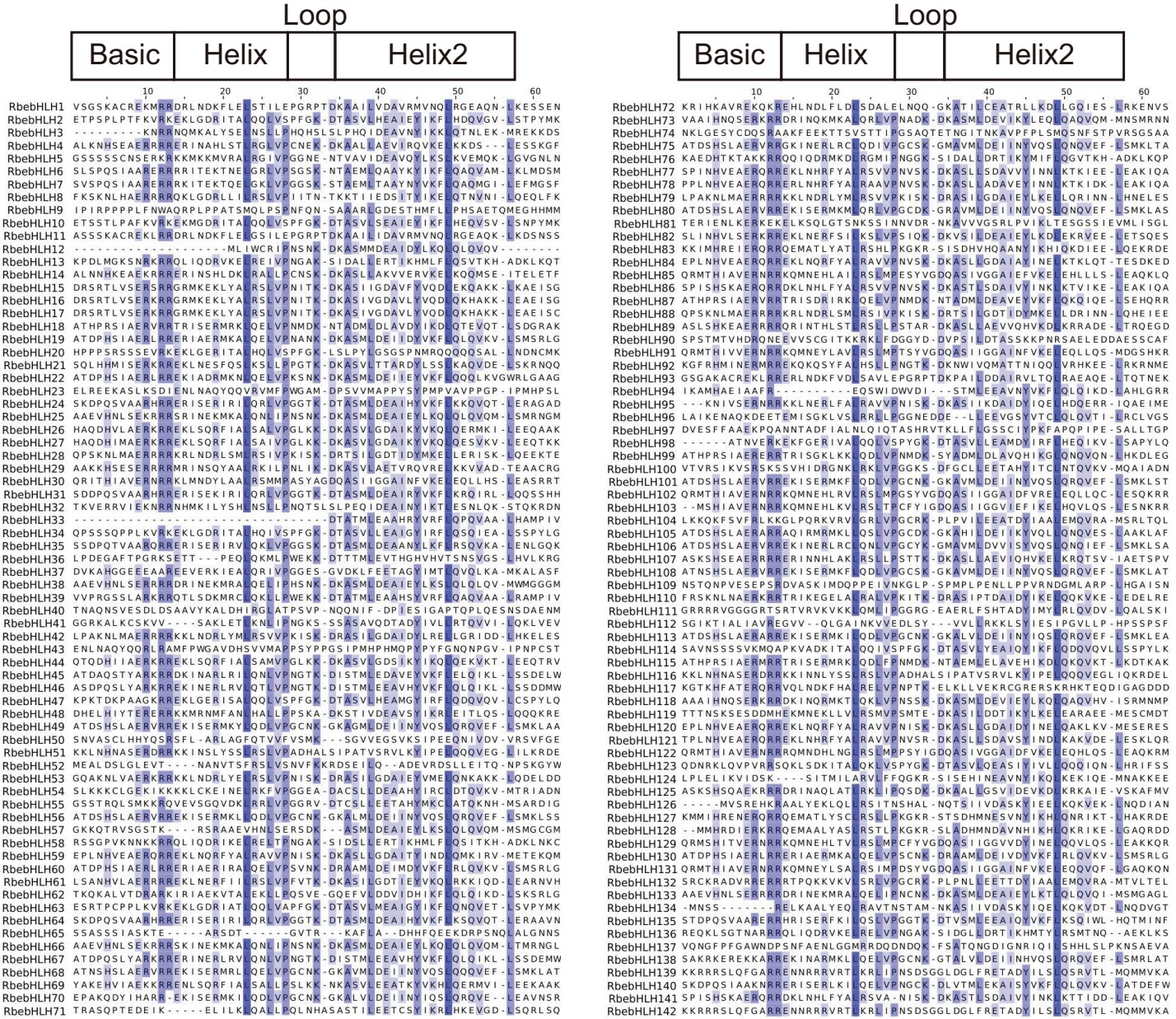

**Figure 1  Multiple sequence alignments of the fasciclin domains of *RbebHLHs*.** The alignment was constructed using MUSCLE and visualized using Jalview. The blue shaded boxes represent conserved amino acid residues. Conserved domains (Basic, HLH) are indicated at the top.

contains two highly conserved sites (Leu23-Leu54). It mainly promotes the interaction between proteins and forms homodimers or heterodimers, which interact with the E-box element in the gene promoter region (*Hernandez et al., 2007*). The conserved domain of the RbebHLH protein obtained by simultaneous artificial construction using MUSCLE is shown in Fig. 1. The Basic domain contained four highly conserved amino acids (H5, E9, R10, R11, and R12), However, the bHLH region contained 18 relatively conserved amino acids (E14, R19, L23, R24, L26, V27, P28, K32, D34, A36, S37, L39, E41, A42, I43, Y45,

K47, L49 and L57), among which R11, R12, L23, and L49 were highly conserved. These residues may play a crucial role in the binding of *bHLHs* to target genes and the formation of protein dimers.

The results of the *RbebHLHs* structural analysis are shown in Fig. 2. Motif prediction by MEME identified 10 conserved motifs. Motif1 and Motif2 were highly conserved in almost all RbebHLH proteins. In addition, each group had some degree of motif specificity. For example, Motif6 only existed in subfamily III and subfamily X, and the combination of "Motif9+Motif10+Motif5" was only detected in subfamily V. The specificity of the motifs may contribute to the functional differences of *bHLHs* in different groups. To clarify the evolution of the *RbebHLHs* family, the bHLH protein sequences of *R. persica* were evaluated by a phylogenetic analysis and exon-intron structure analysis. The RbebHLH proteins were divided into 10 subfamilies. The genes in each subfamily contained similar numbers of exons and introns with relatively conserved positions. The number of exons ranged from 1 (*RbebHLH142, etc.*) to 11 (*RbebHLH123, etc.*), and only about 10% of the genes contained eight or more exons. There were significant differences in the number of introns among different subfamilies. *RbebHLH58* had the largest number of introns, and 18 genes did not contain introns. There were three genes containing only the 5′-UTR, five genes containing only the 3′-UTR, and 31 genes containing both the 5′-UTR and the 3′-UTR. These findings indicate that RbebHLH proteins in the same subfamily generally have a similar composition of conserved motifs, with some variation.

## Phylogenetic and gene duplication analyses

We performed a phylogenetic analysis of bHLH protein sequences from *R. persica* (142 accessions) and *A. thaliana* (162 accessions). Based on the classification of Arabidopsis bHLH proteins (*Heim et al., 2003*), these 304 bHLHs proteins were classified into 26 subfamilies, named from Ia to XV (Fig. 3). We combined some of the similar subfamilies into 21 subfamilies and showed that subfamily XII has the highest number of *RbebHLHs* (24) in *R. persica*, and subfamilies II, IIIe, IX have the lowest number, all of which contain only one *RbebHLH*. It has been shown that proteins belonging to the same subfamily mostly have similar functions, and therefore the phylogenetic results contribute to the function of *RbebHLHs* proteins.

Chromosome mapping revealed that chromosome 6 had the largest number of *RbebHLHs* (26), chromosome 3 had the least (11), and chromosome 5 had the largest number of *RbebHLHs*. There were seven pairs of tandem duplications in *RbebHLHs* (*RbebHLH15* and *RbebHLH16*, *RbebHLH26* and *RbebHLH27*, *RbebHLH52* and *RbebHLH53*, *RbebHLH54* and *RbebHLH55*, *RbebHLH65* and *RbebHLH66*, *RbebHLH111* and *RbebHLH112*, and *RbebHLH127* and *RbebHLH128*) (Fig. S1) and 118 pairs of segmental duplications (Fig. 4). Tandem duplications were present in three pairs on chromosome 6 and in one pair each on chromosome 1, 2, 4, and 5. The most segmental duplications were found on chromosome 2 (32), followed by chromosome 7 (20) (Table S2).

To further explore the evolution of the *bHLH* gene family, we performed a comparative genomic analysis and collinearity analysis among *A. thaliana*, *Prunus. mume*, *Fragaria*

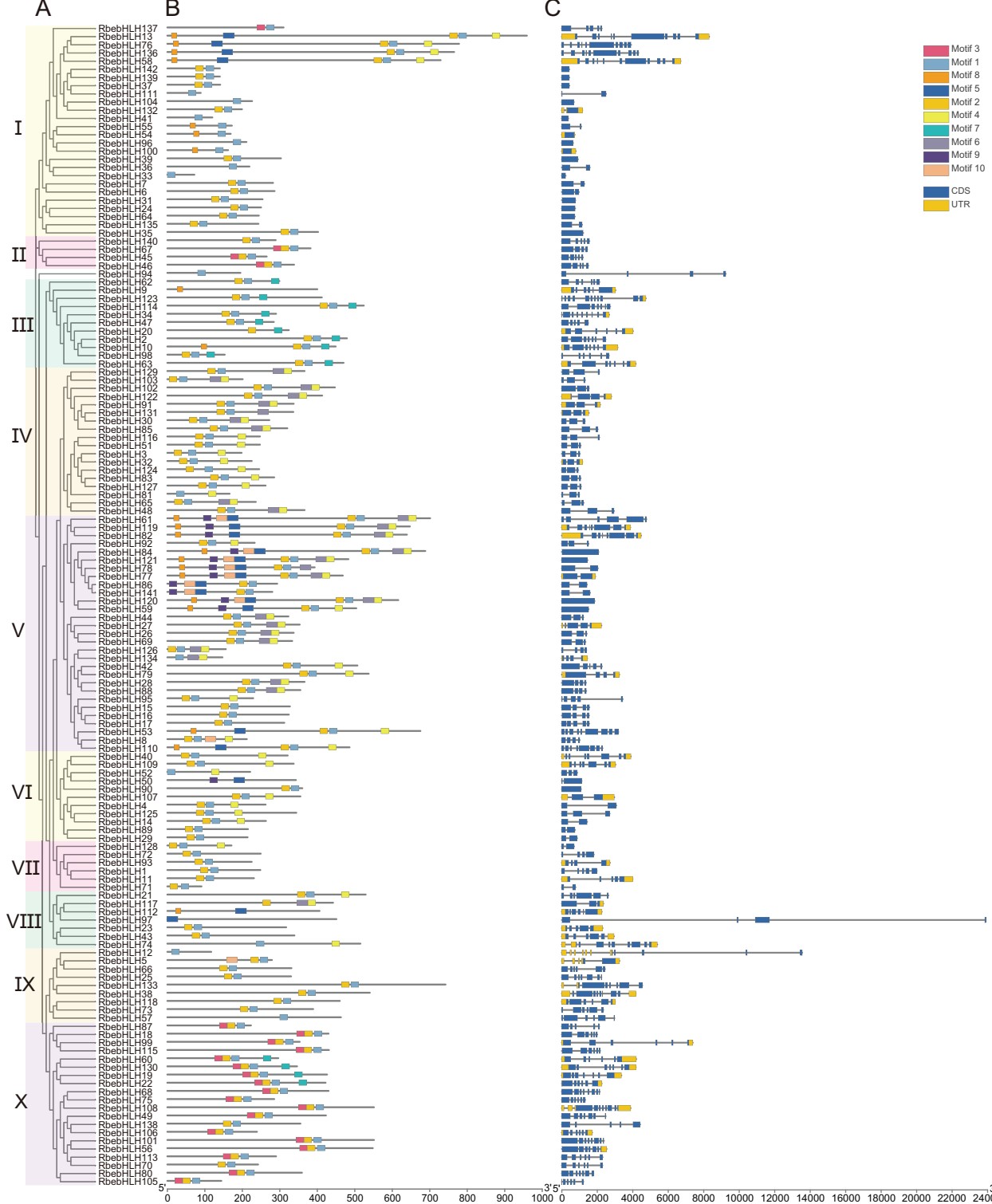

**Figure 2 Phylogenetic relationships, conserved motifs, and exon/intron structures of *RbebHLHs*.** (A) Phylogenetic tree based on the full-length sequences of RbebHLH proteins using the maximum likelihood method and 1,000 bootstrap replicates. (B) *RbebHLHs* were classified into ten groups. The motifs, numbered 1–10, are displayed in different colored boxes. Sequence information for each motif is provided in Fig. S2. (C) Yellow boxes indicate the untranslated region; blue boxes indicate exons; and black lines indicate introns.

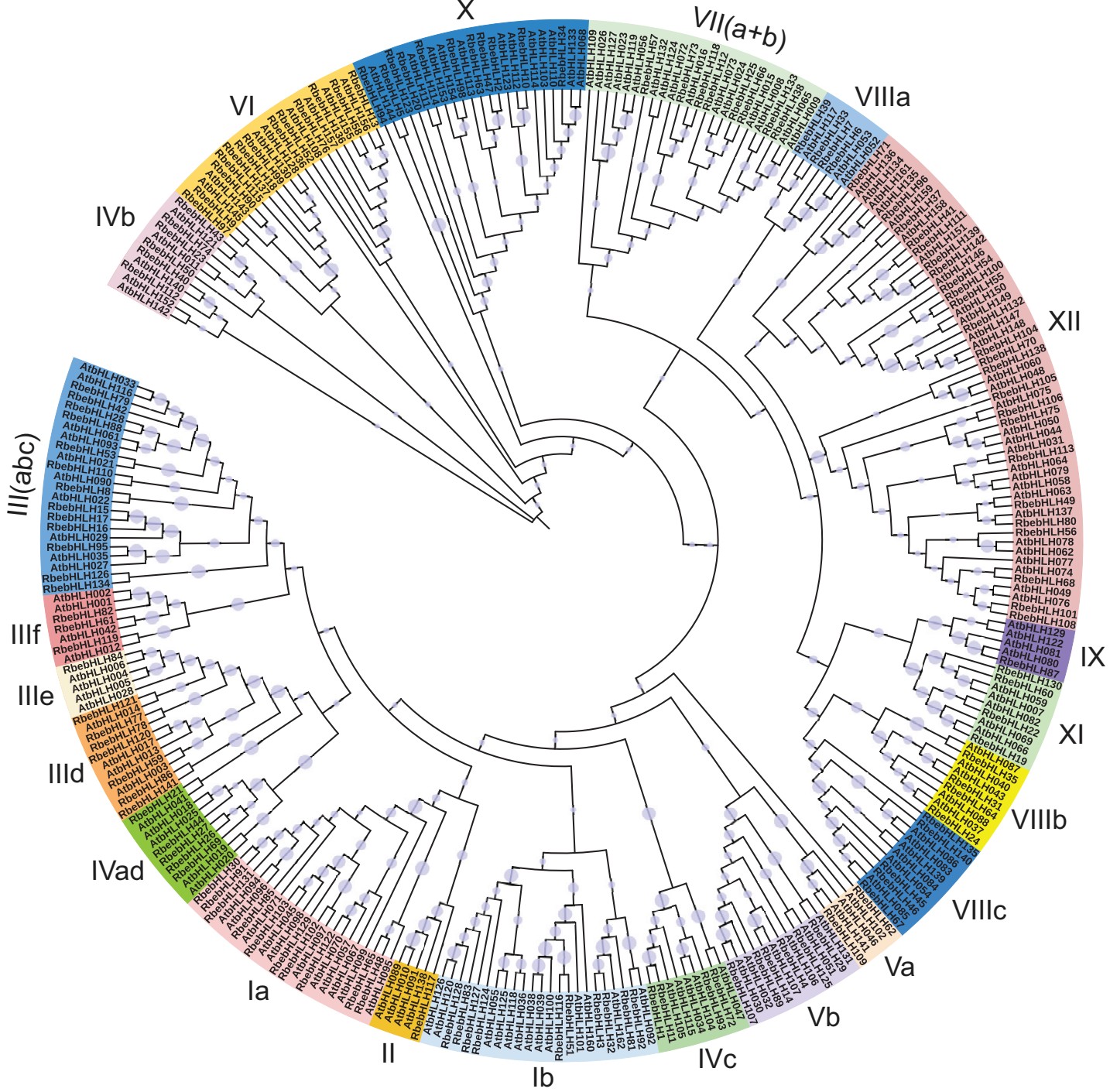

**Figure 3 Phylogenetic tree of *RbebHLHs* and *AtbHLHs*.** All bHLHs were divided into 21 subfamily (Ia–XII) distinguished by color. The circle on the cluster represents the bootstrap support value. The phylogenetic tree was constructed by the neighbor-joining method using MEGA7 with 1,000 bootstrap replicates.

*vesca, Rosa chinensis*, and *Prunus persica* and *R. persica*. There were 127, 150, 151, 172, and 164 pairs of segmental duplications between *R. persica* and *A. thaliana, P. mume, F. vesca, R. chinensis*, and *P. persica*, respectively (Fig. 5). Substantial collinearity was detected

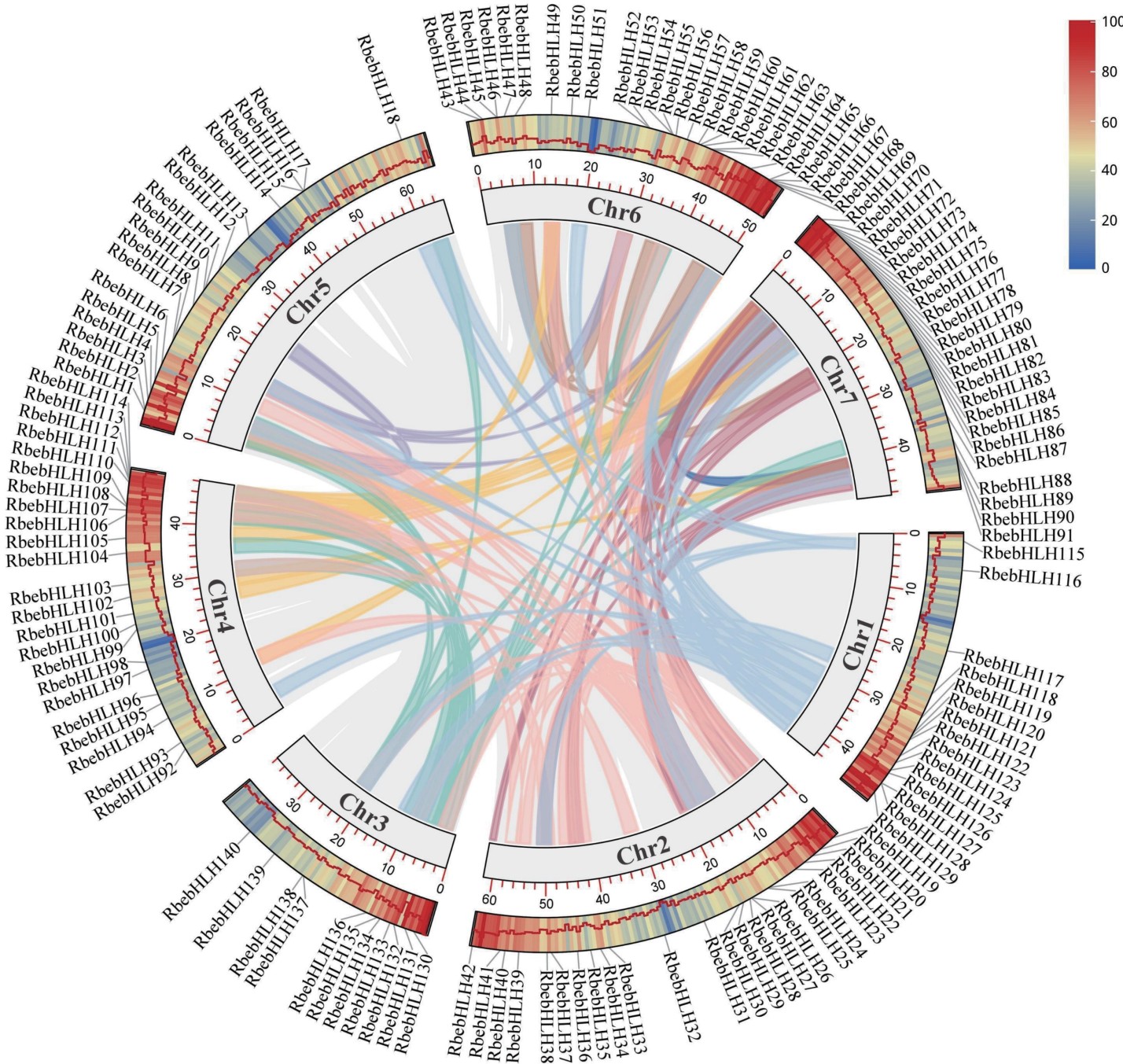

**Figure 4 Location and collinearity relationships of *RbebHLHs*.** Lines in different colors indicate segmental duplications (SD). Boxes indicate chromosomes, and a series of numbers below each box indicate the length of the corresponding chromosome in megabases. The heat map and broken lines indicate the gene density on the chromosome.

between genes in *R. persica* and the other three species in Rosaceae, indicating that they were closely related (Table S3).

To determine the type of selection acting on the collinear genes in *R. persica* and *R. chinensis*, the ratio of the nonsynonymous substitution rate ($K_a$) to the synonymous

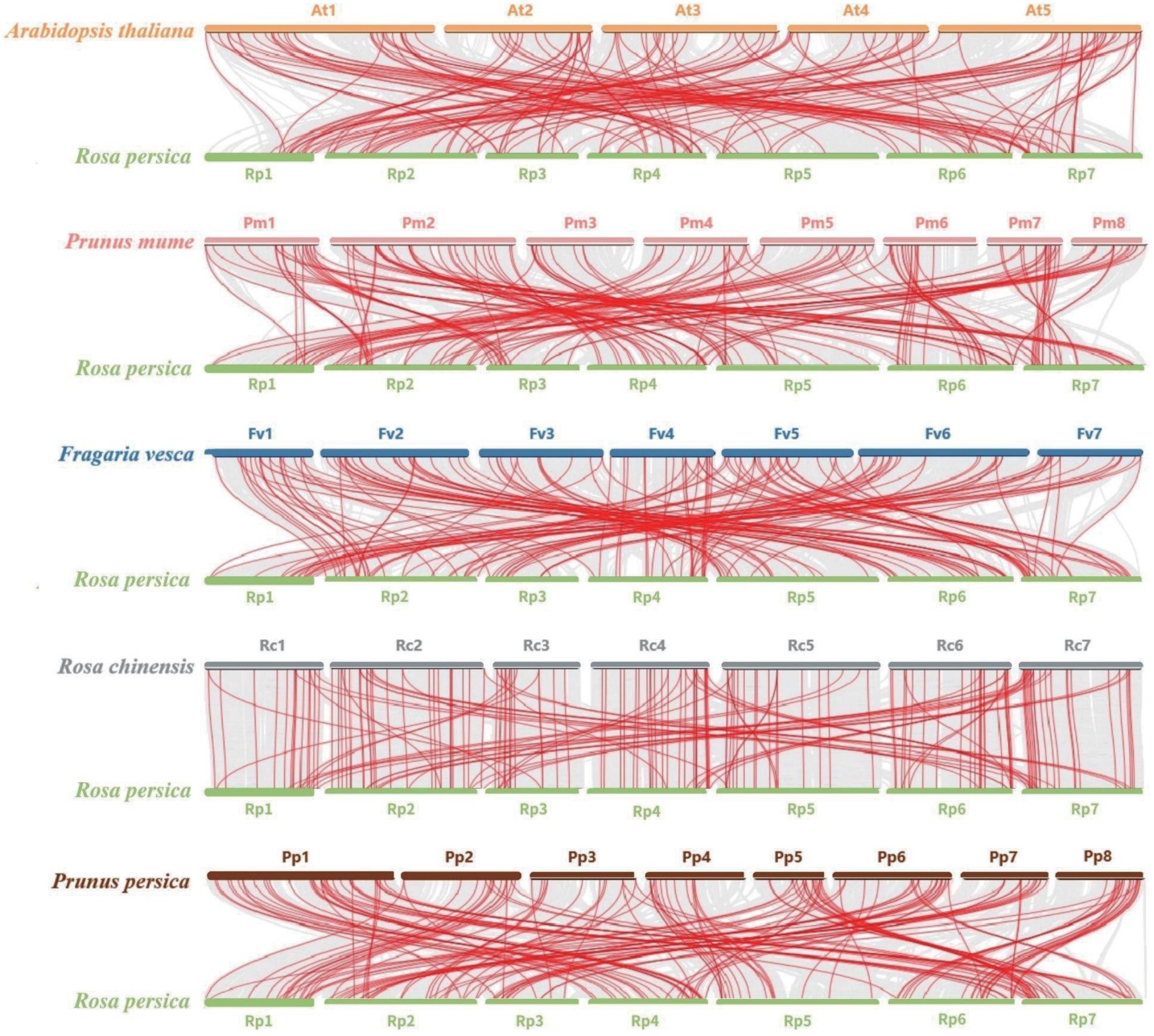

**Figure 5** **Synteny analysis of bHLHs between *A. thaliana*, *P. mume*, *F. vesca*, *R. chinensis*, *P. persica*, and *R. persica*.** Red lines represent colinear bHLH gene pairs, and grey shade highlights syntenic regions between genomes.

substitution rate ($K_s$) was calculated (Table S4). In *R. persica* and *R. chinensis*, only the $K_a$/$K_s$ ratio of *RbebHLH111* and XM_024337436.2 was greater than 1.0 (2.88), suggesting that the gene was under positive selection. The remaining gene pairs had $K_a$/$K_s$ ratios of less than 1.0, consistent with purifying selection. According to the $K_s$ values, the divergence time of *RbebHLHs* and *RcbHLHs* was calculated (Table S4). Divergence time estimates ranged from 1.03 to 274.77 Mya with two distinct peaks, *i.e.*, 115 gene pairs diverged less than 10 Mya and the remaining 44 genes diverged more than 80 Mya.

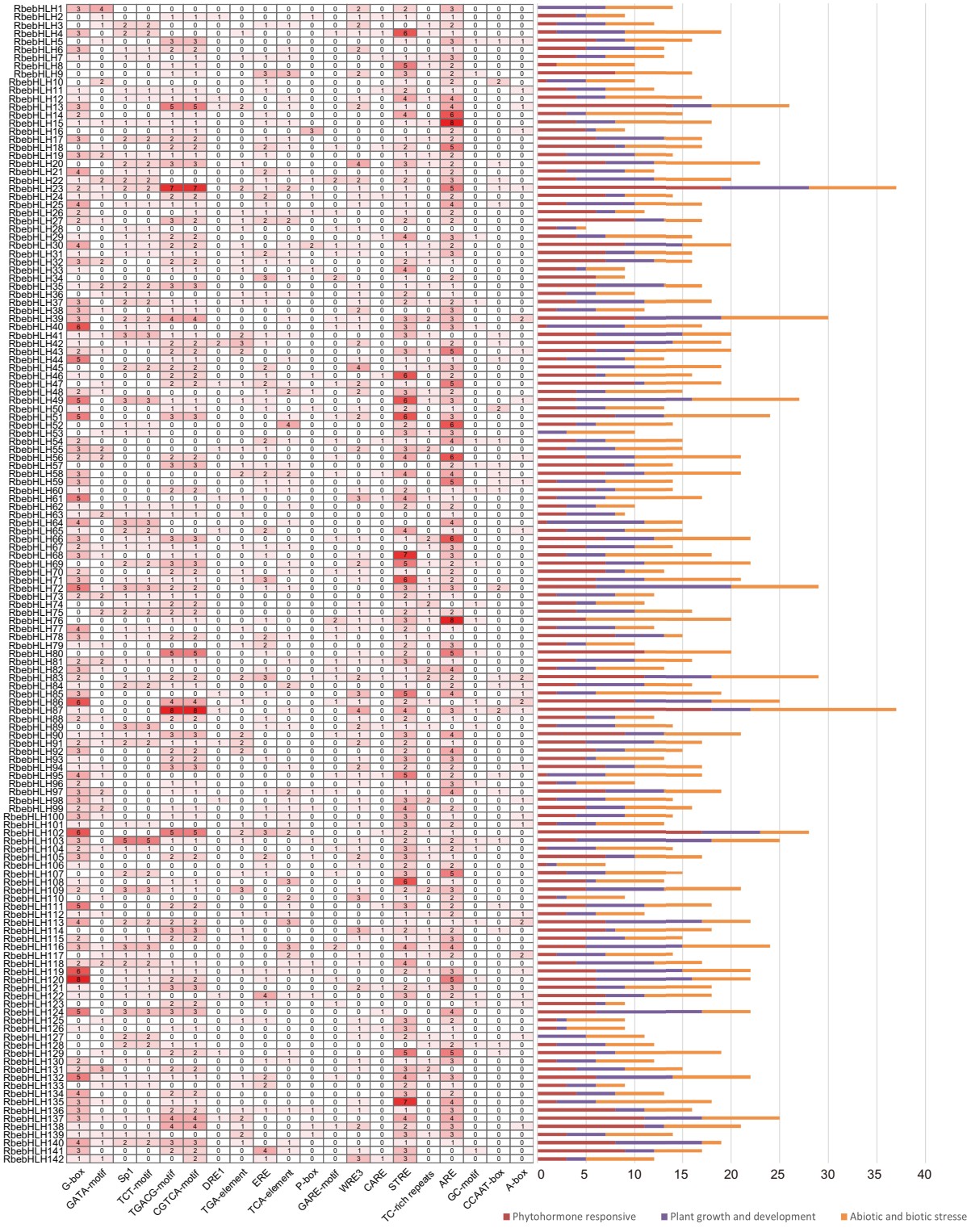

**Figure 6 Analysis of cis-acting elements in the *RbebHLHs* promoter.** The heat map indicates the different cis-acting elements of different *RbebHLHs* promoter regions. The numbers in the boxes indicate the number of occurrences of each cis-acting element in each gene promoter, and the depth of the box color is proportional to the number. The different colors of the right-hand bar graph indicate the number and percentage of cis-acting elements in the three categories (plant growth and development, plant hormone response, and abiotic and biotic stresses).

## Cis-acting element analysis and transcriptional target gene network prediction

To further investigate the potential mechanism by which *RbebHLHs* contribute to the response to abiotic stress, cis-regulatory elements were predicted in the promoter regions. According to functional prediction results, cis-regulatory elements were divided into three categories: environmental stress elements, hormone response elements and development-related elements. The number and type of homeotropic elements for each gene were summarized in a heat map (Fig. 6). *RbebHLH23* and *RbebHLH87* contained the most cis-acting elements (37), while *RbebHLH28* contained the least number of cis-acting elements (two). It can be found that most of the *RbebHLHs* contain both STRE and ARE elements in significantly greater numbers than other cis-acting elements. *RbebHLH76* and *RbebHLH87* contained the largest number of environmental stress-related elements (15), and *RbebHLH28* and *RbebHLH63* contained the fewest (one). In addition, we found that most of the *RbebHLHs* promoter regions have CGTCA-motif and TGACG-motif elements associated with the methyl jasmonate response, suggesting that this part of *RbebHLHs* may respond to low-temperature stress through the jasmonate signaling pathway.These results suggest that *RbebHLHs* are widely involved in various physiological and biochemical activities of plants and in response to environmental stimuli and stresses.

To understand the regulatory relationship between *RbebHLHs* and the *C-repeat binding factors* (*CBF*) family of key cold resistance genes, we used the JASPER database (http://jaspar.genereg.net/) to predict motif information and used MEME FIMO to predict target *CBF* genes (Fig. 7). A total of 21 *RbebHLHs* were predicted to interact with five *CBFs*. The 22 *RbebHLHs* included *phytochrome-interacting factor1* (*PIF1*), *PIF3*, *PIF4*, 3 *SPTs*, 3 *MYC2*, 2 *BIM2*, *DYT1*, *UNE10*, *etc. PIF3*, *PIF4*, 3 *SPTs*, and two *BIM2* interacted with four of the CBFs, and the maximum number of *RbebHLHs* that interacted with both *CBF1* and *CBF4* was 15. These results suggest that a close relationship exists between the *RbebHLHs* and *CBF* families and may regulate the plant response to low-temperature stress through interactions.

## Expression characteristics of selected *RbebHLHs* during overwintering

To understand the response of *RbebHLHs* to low-temperature stress, we selected 30 genes with a high number of cis-acting elements in the promoter region and 21 genes with reciprocal relationships with the CBF family and analyzed their expression patterns during the overwintering period (September-April) (Fig. S3). A total of 45 *RbebHLHs* were analyzed for expression in roots and stems during overwintering, excluding duplicate genes. More *RbebHLHs* were highly expressed in roots in April and in stems in September, while the expression of *RbebHLHs* was generally low in roots and stems in December, January, and February. Both *RbebHLH69* and *Rbe*bHLH86 were not expressed in roots and stems during overwintering, which may be related to the specificity of tissue expression. *RbebHLH8*, *RbebHLH77*, *RbebHLH83*, *RbebHLH121* and *RbebHLH124* showed high expression in the roots in April, whereas they were largely absent in the other months. The expression of *RbebHLH12*, *RbebHLH20*, *RbebHLH23* and *RbebHLH87* gradually

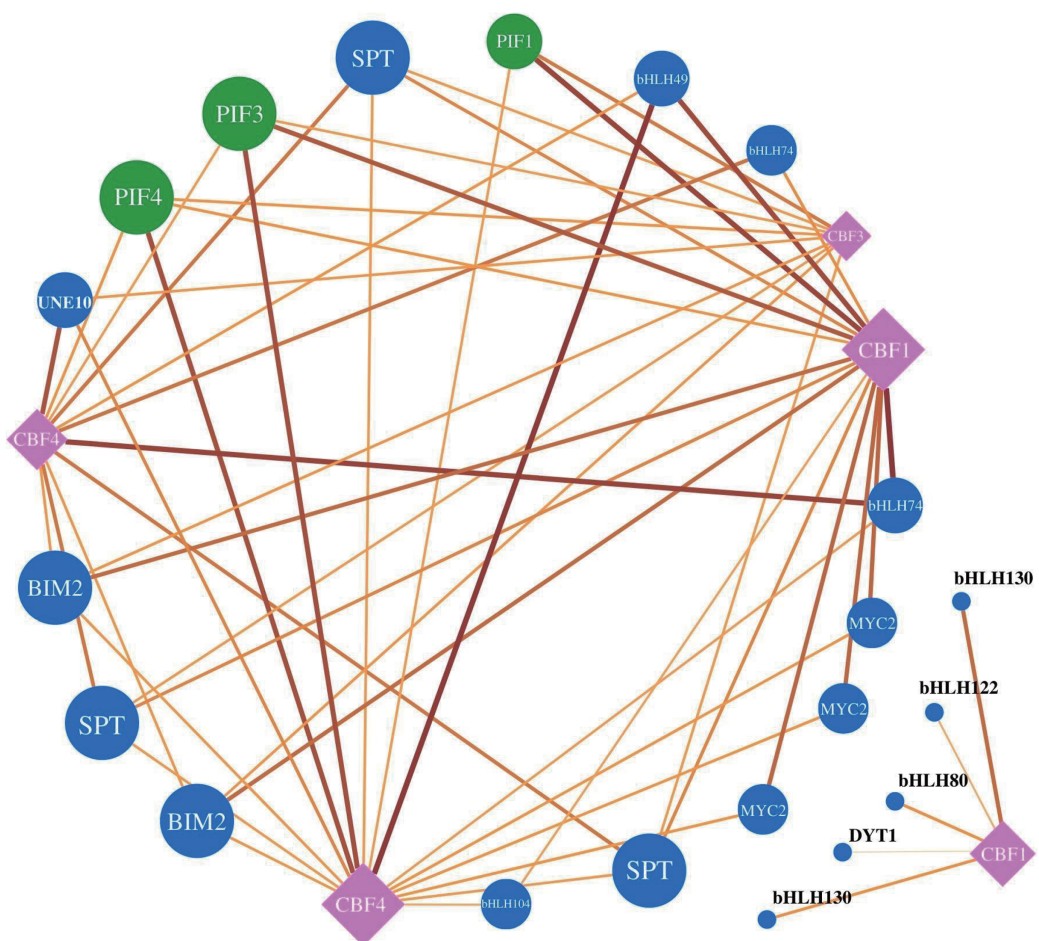

**Figure 7 RbeCBF-RbebHLH target prediction network.** Circles represent the RbebHLH transcription factors, with *RbePIFs* labeled in green and other genes in blue. Squares represent *RbeCBFs*, black lines represent potential regulatory relationships, and the text shows the predicted gene name.

decreased during overwintering, while the expression of most of the selected *RbebHLHs* showed a decreasing and then increasing trend during overwintering. These expression properties suggest that the functions of *RbebHLHs* have diverged to different degrees during evolution.

Given the results of the promoter analysis, we also focused on the expression of genes in the jasmonate signaling pathway with the *RbebHLHs* gene. Expression heatmap analysis revealed that *JAZ6, CBF5, RbebHLH20, RbebHLH87, RbebHLH90* and *RbebHLH130* had similar expression characteristics during overwintering, all with high expression in September. *JAZ8, JAZ10, CBF1, RbebHLH78* and *RbebHLH131* were highly expressed in April (Fig. 8). To verify the accuracy of the transcriptome data, we selected 4 months in the root for qRT-PCR analysis (Table S4). The qRT-PCR results were in general agreement with the transcriptome data, suggesting that *RbebHLHs* may regulate the response of *R. persica* to low-temperature stress through the jasmonic acid signaling pathway.

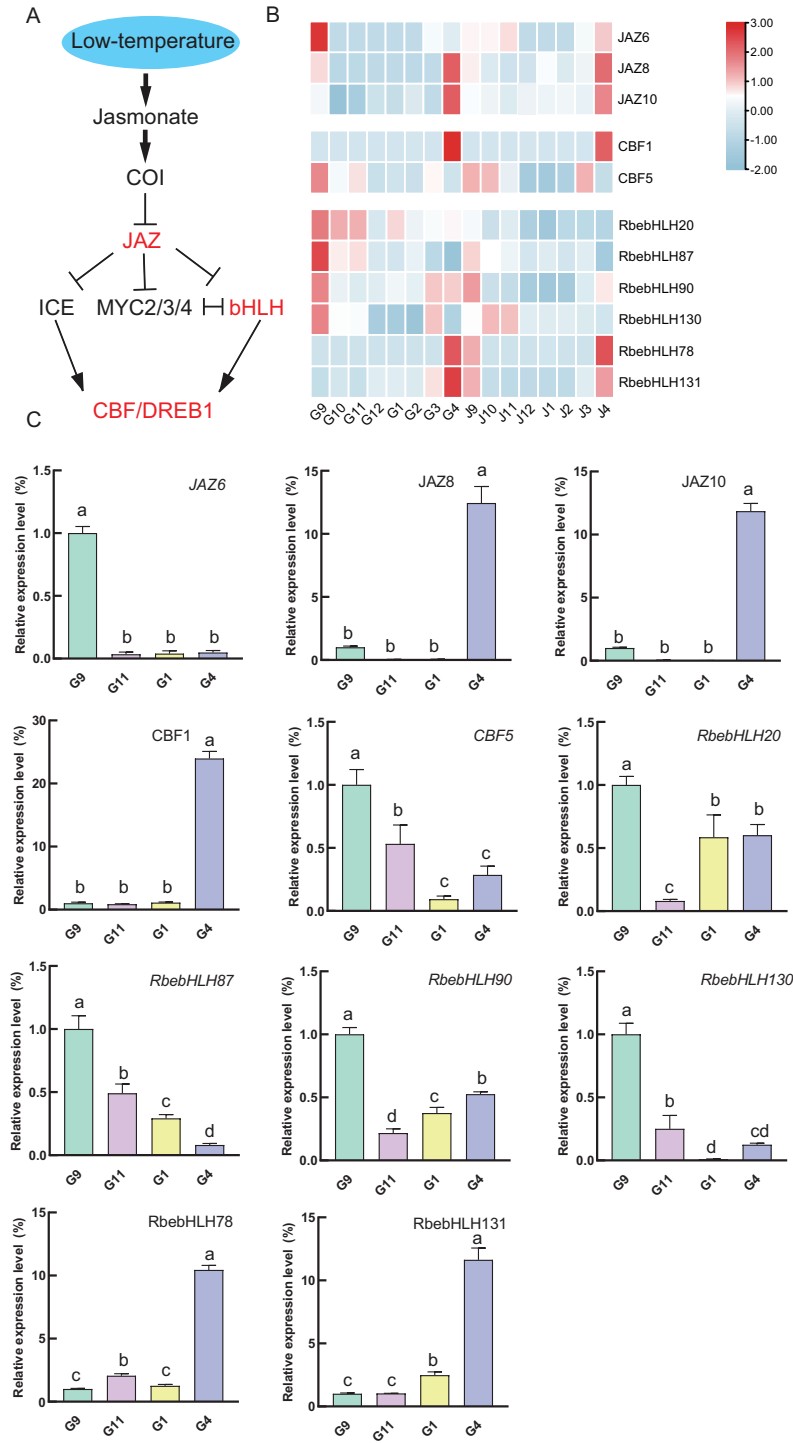

**Figure 8 *RbebHLHs* response to low temperature stress.** (A) Modeling of jasmonic acid-regulated cold signaling pathways. (B) Heatmap of the expression of *JZA*, *CBF* and *RbebHLHs*. G: root; J: stem. Numbers indicate months and colors represent expression levels, with red representing high expression and blue representing low expression. (C) qRT-PCR validation of *JZA*, *CBF* and *RbebHLHs*. The experiments were performed in triplicate using data for September as controls. Lowercase letters indicate *P* < 0.05, as determined using Duncan's test.

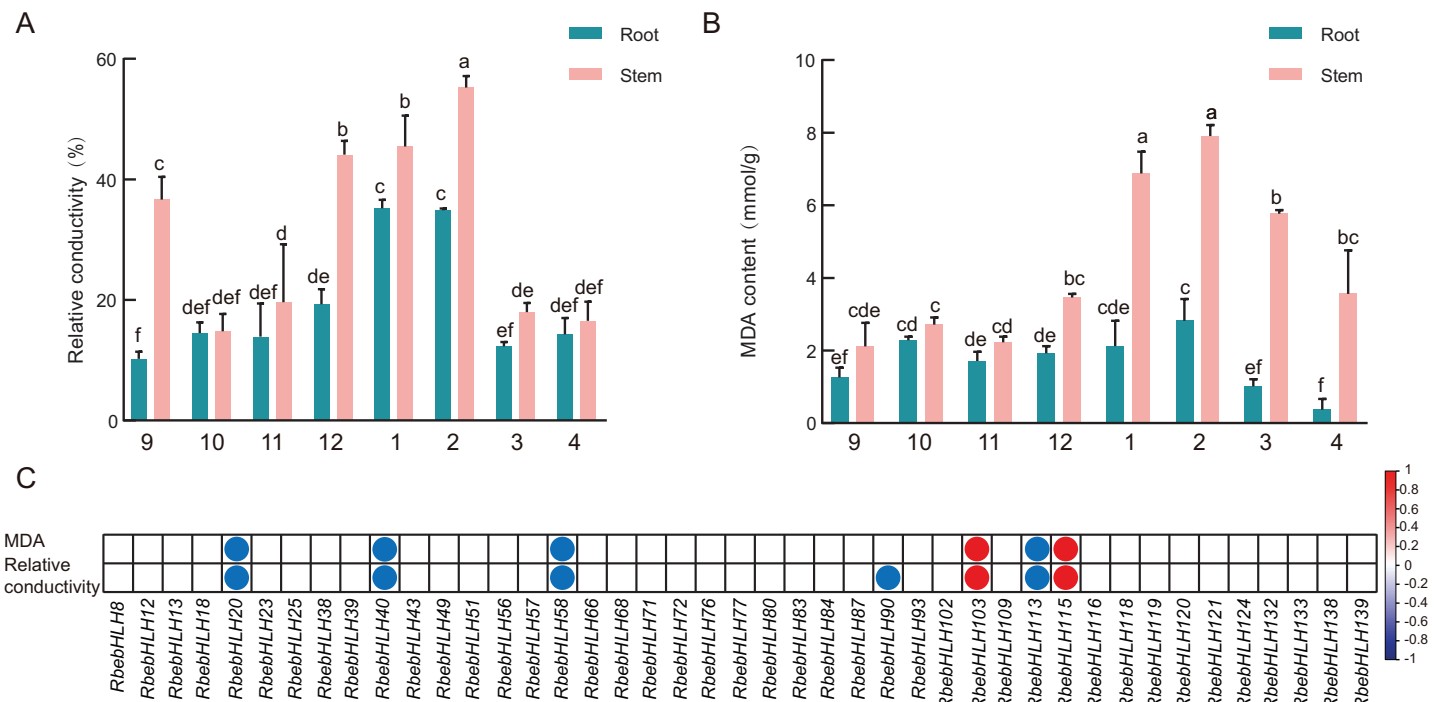

**Figure 9 Relationship between physiological indicators and *RbebHLHs* expression.** Relative electrical conductivity (A) and MDA content (B) of roots and stems during overwintering. The experiments were performed in triplicate using data for September as controls. Lowercase letters indicate $P < 0.05$, as determined using Duncan's test. (C) Correlation of expression of *RbebHLHs* during overwintering with relative conductivity and MDA content. Circles indicate significant correlations ($P < 0.05$) between gene expression levels and physiological indicators. Red circles indicate a positive significant correlation and blue circles indicate a negative significant correlation.               

### Expression of *RbebHLHs* correlates with plant cold tolerance

To investigate the relationship between *RbebHLHs* expression and the cold tolerance of *R. persica*, we measured the relative conductivity and MDA of *R. persica* roots and stems during overwintering (Fig. 9). Relative conductivity and malondialdehyde content are indicators of cell membrane integrity and lipid peroxidation, respectively.

The conductivity of both roots and stems showed a trend of increasing and then decreasing, with the lowest conductivity in September and a peak in January for roots. Stem conductivity was lowest in October and peaked in February. The malondialdehyde content also showed a trend of increasing and then decreasing, with the lowest malondialdehyde content in the root part in April and the peak in February. The relative conductivity was higher in December, January and February than in other months, while malondialdehyde content was higher in January, February and March. This indicates that *R. persica* was more severely affected by low-temperature stress in these months. Correlation analysis of the expression of the selected 45 *RbebHLHs* with relative conductivity and malondialdehyde content showed that six genes were significantly correlated with malondialdehyde content and seven genes were significantly correlated with relative conductivity (Fig. 9C). Among them, *RbebHLH20*, *RbebHLH40*, *RbebHLH58* and *RbebHLH113* were significantly negatively correlated with both malondialdehyde and relative conductivity, while *RbebHLH130* and *RbebHLH115* were significantly negatively

correlated. These results suggest that *RbebHLHs* play an important role in regulating *R. persica* in response to low-temperature stress.

## DISCUSSION

Bioinformatics methods have been used to analyze the evolution and function of *bHLHs* in a variety of plant taxa. The *bHLHs* in animals are categorized into subgroups A-F based on their structure, and many of the plant *bHLHs* identified to date are closely related to subgroup B (*Sailsbery & Dean, 2012*). Initially, the 133 *bHLHs* in Arabidopsis were categorized into 12 subclades. Subsequently, 14 new *bHLHs* were identified and further classified into 21 subclades, but this classification was limited to higher land plants (*Heim et al., 2003*; *Toledo-Ortiz, Huq & Quail, 2003*; *Zhang et al., 2020*). As research has progressed, studies including mosses, algae, and angiosperms have identified approximately 33 distinct bHLH subgroups (*Carretero-paulet et al., 2010*). Plant *bHLHs* are diverse in number and taxonomy. A total of 161, 167, and 152 *bHLHs* were identified in Arabidopsis (*Arabidopsis thaliana*), rice (*Oryza sativa*), and tomato (*Solanum lycopersicum*), respectively, which were classified into 21, 22, and 24 subgroups, respectively (*Sun, Fan & Ling, 2015*). However, little is known about the members and expression profiles of *bHLHs* in *Rosa persica* owing to the restricted distribution of the species, among other reasons. In this study, 142 *RbebHLHs* were identified in the *R. persica* genome, followed by analyses of phylogenetic relationships, gene structure, collinearity, gene duplications, and expression patterns. Based on a phylogenetic analysis, gene structure, and conserved motifs, 142 *RbebHLHs* could be divided into 10 subclasses. Gene structure and conserved motif analysis revealed that the gene structure and conserved motif distribution of *bHLHs* in the same subclass were basically similar, whereas members of different subclasses differed significantly in motif composition (Fig. 2). This may be due to significant changes in the amino acid composition outside the conserved structural domains of bHLH proteins as a result of various events (*Pires & Dolan, 2010*), such as the uneven amplification of gene families, mixing of internal domains, and splicing of introns (*Carretero-paulet et al., 2010*). This variation corresponds to functional differences among bHLH proteins. The results of chromosome localization showed that the number of genes was highest on chromosome 6 rather than on chromosome 5, which is the longest in length, indicating that the number of genes distributed is independent of chromosome length.

According to previous studies, *bHLH* functions are mainly classified into the following categories: plant growth, development, and morphogenesis, plant stress resistance, and plant secondary metabolism (*Li, 2017a*). We focus on the role of *bHLH* in stress resistance in *R. persica*. Identification of cis-acting elements in stress-responsive promoters is important for understanding the molecular switching of stress-induced genes (*Yamaguchi-Shinozaki & Shinozaki, 2005*). Methyl jasmonate (MeJA), as a hormone involved in plant signaling, can alleviate various stresses and counteract the toxicity of pathogenic bacteria, salt stress, drought stress, low-temperature stress, and heavy metal stress (*Yu et al., 2019*). In addition, the *bHLH* transcription factor *MYC2* is a major regulator of jasmonate signaling and is involved in abiotic stress-regulated signaling pathways (*Wang et al.,*

2022b). A study of the 2,000 kb promoter region upstream of *bHLH* found that the Methyl jasmonate response-associated CGTCA-motif and TGACG-motif, and the stress-response-associated STRE and ARE are present in most *RbebHLHs* (Fig. 6). Thus, it is likely that *RbebHLHs* responds to abiotic stress through the jasmonate pathway. Low-temperature stress presents a serious threat to plant survival and reproduction. Previous studies of cold resistance mechanisms have found that the ICE-CBF-COR pathway plays an important role in improving plant cold tolerance (*Jiang et al., 2020*). AtICE1 (AtbHLH116) contains a bHLH domain, encoding a transcription activator gene (MYC)-like bHLH transcription factor, which specifically binds to the MYC action element in the CBF promoter at low temperatures to induce the expression of CBF and promote the expression of a series of downstream cold-induced genes, thereby improving the cold resistance of transgenic plants (*Chinnusamy et al., 2003*). The overexpression of the bHLH transcription factor *DlICE1* of *Dimocarpus longan* in Arabidopsis can increase the proline content, reduce the ion extravasation rate, and reduce malondialdehyde (MDA) and reactive oxygen species (ROS) accumulation, thereby improving plant cold tolerance (*Yang et al., 2019*). Under low-temperature stress, the expression levels of *AtCBF1, AtCBF2,* and *AtCBF3* in the *ICE1-CBF* pathway and cold response genes (*AtRD29A, AtCOR15A, AtCOR47,* and *AtKIN1*) in the overexpression lines were also significantly higher than those in wild-type plants (*Liu et al., 2015*). Previous studies have proven that *PIF1, PIF3,* and *PIF4* negatively regulate *CBF* expression and freezing tolerance (*Ding, Shi & Yang, 2020*). *PIF4* is a key regulator in plant thermal morphogenesis, which is mediated by plant hormones, the biological clock, and light (*Quint et al., 2016*). The overexpression of *TaMYC2A*, B, and D increased the expression level of Inducer of *CBF* expression (ICE)-CBF-Cold-regulated (COR) in polar Arabidopsis under extremely cold conditions, thereby improving its freezing resistance (*Wang et al., 2022b*). In this study, 21 *RbebHLHs* were found to have possible interactions with different CBFs, suggesting that *RbebHLHs* genes play an important role in *R. persica* response to low-temperature stress. Our analysis of the expression of jasmonate signaling pathway genes during overwintering revealed that some of the *RbebHLHs* genes may be regulated by JAZ and respond to low-temperature stress through the jasmonate signaling pathway and the ICE1-CBF signaling pathway (Fig. 8).

The degree of tolerance to low-temperature stress varies considerably among different life types, with woody plants often showing greater cold tolerance in order to adapt to seasonal changes in the growing environment, especially to the effects of low-temperature stress in winter. The more commonly used methods for evaluating the cold hardiness of woody plants at this stage include differential thermal analysis (DTA), electrolyte leakage (EL), and physiological and biochemical index measurements (*Wang et al., 2022a*). We utilized naturally overwintered outdoor *R. persica* material to determine the electrolyte exudate rate and MDA content, which were combined with the expression of *RbebHLHs* to analyze the role of *RbebHLHs* in the response of *R. persica* to low-temperature stress. Expression analysis of *RbebHLHs* showed that most of the *RbebHLHs* had lower expression in the coldest month (December–February) than in the other months, while the opposite was true for *RbebHLH39, RbebHLH76, RbebHLH102,* and *RbebHLH115* (Fig. S3).

Correlation analysis showed that the expression of *RbebHLH40* was significantly negatively correlated with relative conductivity and MDA content, and the expression of *RbebHLH115* was significantly positively correlated with relative conductivity and MDA content (Fig. 9). In addition, *RbebHLH40* (*BIM2*) and *RbebHLH115* (*RbebHLH130*) have potential interactions with the CBF family, so we hypothesized that *RbebHLH40* positively regulates R tolerance to low-temperature stress, whereas *RbebHLH115* negatively regulates *R. persica* tolerance to low-temperature stress, and both genes act through the *ICE-CBF-COR* pathway.

## CONCLUSIONS

In this study, 142 *RbebHLHs* be divided into 21 subclades based on the *R. persica* genome data. The bHLH protein sequences in the same subfamily shared a similar conserved motif composition. An analysis of the $K_a/K_s$ ratio indicated that *RbebHLHs* underwent strong purifying selection. The expression of *RbebHLHs* during overwintering was analyzed in conjunction with promoter and *CBF* interaction regulatory networks. Expression analysis implied that part of bHLH may respond to low-temperature stress in *R. persica* through the jasmonate and *ICE-CBF* pathways. Correlation analysis of *RbebHLHs* with relative conductivity and MDA content demonstrated that some of the *RbebHLHs* play a cold acclimatization role during natural *R. persica* overwintering. Therefore, this study deepens our understanding of *RbebHLHs* and provides a theoretical basis for screening *R. persica* stress tolerance genes.

### Funding

This work was supported by the National Natural Science Foundation of China (Grant No. 32071818) and Beijing High-Precision Discipline Project, Discipline of Ecological Environment of Urban and Rural Human Settlements. The funders had no role in study design, data collection and analysis, decision to publish, or preparation of the manuscript.

### Grant Disclosures

The following grant information was disclosed by the authors:
National Natural Science Foundation: 32071818.
Beijing High-Precision Discipline Project, Discipline of Ecological Environment of Urban and Rural Human Settlements.

### Competing Interests

The authors declare that they have no competing interests.

### Author Contributions

- Yueying Zhuang conceived and designed the experiments, performed the experiments, analyzed the data, prepared figures and/or tables, and approved the final draft.
- Lijun Zhou conceived and designed the experiments, performed the experiments, analyzed the data, prepared figures and/or tables, and approved the final draft.
- Lifang Geng conceived and designed the experiments, performed the experiments, analyzed the data, prepared figures and/or tables, and approved the final draft.
- Lv Jiang conceived and designed the experiments, performed the experiments, analyzed the data, prepared figures and/or tables, and approved the final draft.
- Yunji Sui conceived and designed the experiments, authored or reviewed drafts of the article, and approved the final draft.
- Le Luo conceived and designed the experiments, authored or reviewed drafts of the article, and approved the final draft.
- Huitang Pan conceived and designed the experiments, authored or reviewed drafts of the article, and approved the final draft.
- Qixiang Zhang conceived and designed the experiments, authored or reviewed drafts of the article, and approved the final draft.
- Chao Yu conceived and designed the experiments, authored or reviewed drafts of the article, and approved the final draft.

### Data Availability

  The Rbebhlh cds sequences are available at GenBank: OQ544945 to OQ545086.
The raw data for qRT-PCR (Fig. 7) is available in the Supplemental File.

### Supplemental Information

Supplemental information for this article can be found online at http://dx.doi.org/10.7717/peerj.16568#supplemental-information.

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
