# Peer review of "Genome-wide identification of the bHLH transcription factor family in Rosa persica and response to low-temperature stress"

_PeerJ, doi:10.7717/peerj.16568_

## Round 0.1 · original submission · Major Revisions

Based on the reviewer's comments and my own assessment, this manuscript can be re-accepted after rigorously following and incorporating the flaws/missing information as raised by reviewers.

In this manuscript, the authors have used bioinformatics tools to systematically identify the bHLH gene family in Rosa persica, including analysis of structural domains, promoter regions, and collinearity. However, the manuscript has the following major issues:

1、The manuscript lacks a clear scientific question and provides only descriptive analyses of this gene family. There is no final conclusion drawn from this analysis.

2、The logical flow of the manuscript is problematic. For instance, the authors identified candidate key genes based on promoter element analysis, but failed to connect these results to the subsequent expression analysis under low-temperature treatment. This lack of coherence makes it difficult to draw meaningful conclusions.

3、The authors did not include a rigorous control experiment in the field study of the effect of cold stress on gene expression. The use of samples collected from different months in the wild does not control for other factors besides temperature that may affect gene expression. I recommend that the authors add a rigorous control experiment to strengthen their results.

Overall, while the authors have conducted a comprehensive bioinformatics analysis of the bHLH gene family in Rosa persica, the manuscript requires significant revisions to address the issues raised above.

·

Basic reporting

The manuscript of Zhuang et al. presents the identification about the roles and expression patterns of bHLH family genes in Rosa persica. The authors systematically analyzed the RbebHLHs by bioinformatics, and their expression patterns in different tissues and under low temperature stress were analyzed by transcriptome and qRT-PCR.
Although, the work is valuable but requires a few corrections before being published.

- Major concerns:
1. Line 19 –“ …responses and have been studied…”. The word “and”, whether to delete?

2. Lines 22-23, “This study systematically identified members of the RbebHLH gene family based on genome data.” Whether to be deleted this sentence?

3. Lines 26-27, this sentence feels a little redundance.

4. Line 175, isoelectric point should be abbreviated to pI.

5. Line 171, it might be more informative to rename " RbebHLHx" to something more descriptive, such as the ortholog about Arabidopsis. Based on the present nomenclature, it is difficult to see which RbebHLH is most similar to AtbHLH.

6. Lines 181-186, This part of the content placed in the results section is questionable.

7. Line 198, The phylogenetic trees of Fig.2 were used their DNA or protein sequences, the authors did not point out it. Is it more appropriate for the phylogenetic tree in Figure 2 to be constructed by their cDNA sequence, since this part mainly involves with gene structure analysis?

8. Color of CDS and UTR in Fig.2C is difficult to distinguish, please change other colors!

9. Lines 216-220, the bHLHs proteins were divided into 21 subfamilies, called subfamilies A-U. Whether the authors can assign the name according to the reported function, such as Clade MYC, PIF, BEE, et al. (Jiang et al 2022, PMID: 35218399; Heim et al 2003), which could provide clues about their roles. Because the function of orthologous genes is usually conserved, so we can get a rough deduce of their function.

10. Line 268, the word “SPTS” should be “SPTs”.

11. Lines 269-278, these contents are more appropriate in the Discussion section.

12. Line 279, this section also contains gene expression at different developmental stage, so the subheading needs to improve. In addition, the expression of different tissues and different developmental stages is best described separately as good. Moreover, Fig. 7B should be placed in this section.

13. Line 281, “root, stem, leaf, flower, and fruit” should be plural.

14. Line 282, “lilobase”?

15. Line 386, RbebHLHs be divided into 21 subclades based on the R. persica genome data,

16. Moreover, the writing format of manuscript need to be checked carefully, such as the format for Latin name of plant need to add the full name at the first appears in the manuscript, and other spell errors. It needs to be revised.

17. Unfortunately, there are lots of mistakes in part of reference, such as R10, R11, R12, R14, R17, R20, R21, R23, R24, R31, R33, R36, R38, R39, R42, R44, R46, R49, R5, and so on. Mistakes types included short of pages, italic of gene name and Latin name, capital letters of title words and journal name, the format of references need to be uniform all of them. Please revise them carefully.

Experimental design

no comment

Validity of the findings

no comment

Reviewer 2 ·

Basic reporting

1、The language of this manuscript needs to be further improved and polished.
2、Literature references are sufficient.
3、The structure of the article is complete, but there are many figures that are quite disorganized, such as A and B in Figure 7 being mistakenly labeled. Please ask the author to carefully check them. The original data of the transcriptome used in the article has not been provided.

Experimental design

1、The scientific questions in this article are not clear, the logical flow is poor, and the problem is not well-defined.
2、There are some unclear research methods in this article or other methods that could be attempted. For example, in lines 88-90, the author used the BLAST method to identify candidate bHLH genes. However, there is a corresponding PFAM file for this family in Pfam. Why was the more accurate and commonly used HMMER not used for identification? In lines 91-92, the candidate genes were further screened using the bHLH domain, but you introduced two conservative domains in the background. Why was only one domain used for screening? The process of transcriptome analysis was not clearly explained in lines 141-144, and the raw data was not provided.

Validity of the findings

This article is only an informational manuscript and lacks significant innovation. It does not propose clear scientific questions and the logic of the article is not clear.

Additional comments

Dear Editor,

I have reviewed the manuscript titled "Genome-wide identification and expression analysis of the bHLH transcription factor family in Rosa persica" and would like to provide my feedback on this work.

The authors have used bioinformatics tools to systematically identify the bHLH gene family in Rosa persica, including analysis of structural domains, promoter regions, and collinearity. However, the manuscript has the following major issues:

1、The manuscript lacks a clear scientific question and provides only descriptive analyses of this gene family. There is no final conclusion drawn from this analysis.
2、The logical flow of the manuscript is problematic. For instance, the authors identified candidate key genes based on promoter element analysis, but failed to connect these results to the subsequent expression analysis under low-temperature treatment. This lack of coherence makes it difficult to draw meaningful conclusions.

3、The authors did not include a rigorous control experiment in the field study of the effect of cold stress on gene expression. The use of samples collected from different months in the wild does not control for other factors besides temperature that may affect gene expression. I recommend that the authors add a rigorous control experiment to strengthen their results.

Overall, while the authors have conducted a comprehensive bioinformatics analysis of the bHLH gene family in Rosa persica, the manuscript requires significant revisions to address the issues raised above.

Reviewer 3 ·

Basic reporting

Zhuang et al discusses the identification and expression analysis of bHLH transcription factor family in Rosa persica. Authors have performed gene structure, phylogenomic analysis, motif analysis, structure analysis, and gene expression analysis to speculate the functions of bHLH gene family. The manuscript has explored the characteristics of bHLH genes through sequence alignment and other bioinformatics tools. It limits the scope of study as the manuscript is not clear about the role of bHLHs in stress resistance and flower spot formation in Rosa persica.

Introduction:
1. In the result section authors discuss the aim of the manuscript to elucidate bHLH role under low temperature (line 27). However, in the introduction section, author mentioned the purpose of manuscript is to provide a reference about bHLHs in stress (what stress is also not clear) and flower spots. It contradicts the purpose of the manuscript. It should be rewritten, and focus should be clear on whether authors would like to keep manuscript in general or would like to explore key bHLHs during a specific process.
2. What is the criteria authors used to differentiate between the DNA binding domain (DBD), helical region, and dimerization domain? Further, authors showed the second helical region ended at 49th position towards C-terminal, however, the sequence alignment shows the helix is not ended at that position and a presence of Leucine (L) in the flanking heptad.
3. The DBD of bHLH3, bHLH12, and bHLH33 is not visible. It's significance?
4. Line 195 discusses the conserved L but do-not mention the L conserved at 57th position.
5. What is the logical explanation for close proximity of bHLH 137 and bHLH13? No clear DBD signature sequence present in the bHLH137 (figure 1) as compared to the bHLH13. Further is there any criteria for the c-terminal boundary for bHLHs like bZIPs (Deppmann et al. 2004 and Jain et al., 2017)?
6. In figure 2, why do authors use genomic sequences for the classification instead of protein sequences, which would be more relevant as bHLHs act as homodimers or heterodimers?
7. In figure 3, color coding for B and C must change.
8. I am still wondering what genomic database authors have used for genomic alignment, and phylogenetic analysis. Also, can the authors share the details of reference paper, which discusses the denovo sequencing of Rosa persica.
9. Supplementary figure S3 and the analysis related to the promoter region of bHLHs are not clear to me. bHLHs are transcription factors, and their activity usually lies with their homo- or heterotypic interactions and their binding to target promoter regions instead of their own promoter regions. Further, the promoter region of all bHLHs have numerous binding motifs, so how can we define the specificity of bHLH to a particular stress condition?
10. Why was the c-repeat binding factor chosen in the current study and no other resistance or flower spot-related genes (line 80)?
11. In figure 7 C, color coding is not clear. No details are mentioned related to months in the legends. It is also not clear the significance of expression data of selected bHLHs.


Minor

1. References missing for the accession of R. persica and A. thaliana used for phylogenetic analysis.


References
1. Deppmann, C.D., Acharya, A., Rishi, V., Wobbes, B., Smeekens, S., Taparowsky, E.J. and Vinson, C., 2004. Dimerization specificity of all 67 B-ZIP motifs in Arabidopsis thaliana: a comparison to Homo sapiens B-ZIP motifs. Nucleic acids research, 32(11), pp.3435-3445.
2. Jain, P., Shah, K., Sharma, N., Kaur, R., Singh, J., Vinson, C. and Rishi, V., 2017. A-ZIP53, a dominant negative reveals the molecular mechanism of heterodimerization between bZIP53, bZIP10 and bZIP25 involved in Arabidopsis seed maturation. Scientific Reports, 7(1), p.14343.
1.

Experimental design

NA

Validity of the findings

NA

Additional comments

NA

---

## Round 0.2 · Minor Revisions

This revised manuscript looks much better with substantial improvements with valuable insights. Authors are advised to follow the minor comments by Reviewer 1 and try to incorporate the mechanistic overview of bHLHs under low temperatures as requested by Reviewer 3. This can be included in the discussion section and you are also advised to add more conclusive information in the last section of this manuscript.

·

Basic reporting

Thank you for your efforts in addressing my comments and revising your manuscript. There are a couple additional minor comments identified in the review of your revised manuscript. I hope you are willing to make these minor changes. I recommend the present version of the manuscript to PeerJ for acceptance after minor changes.

1. Many errors in literature need to check and correct. For example,
L489, Ipomoea purpurea should be italic.
L524, pages?
L550, the literature starts with a new line.
The pages of literature should be on the same line.
The DOI format needs to be uniform. Is it DOI or https://doi.org/?
2. The origin and evolution of bHLHs or subfamilies (PIF, MYC, ILR3) in plant species and even algae should be discussed.

Experimental design

No comments

Validity of the findings

No comments

Additional comments

No comments

Reviewer 2 ·

Basic reporting

'no comment

Experimental design

'no comment

Validity of the findings

no comment

Reviewer 3 ·

Basic reporting

The authors did the substantial improvements in the manuscript; however, the manuscript still unable to answer the basic questions they posed about the valuable insights and potential mechanisms of bHLHs under low temperature (as seen in lines 37 and 79). The data discussed by the authors consists solely of a genome-wide alignment and distribution of bHLHs under low temperature in Rosa persica, rather than providing a mechanistic interpretation and identification of key bHLHs under low temperature.

Experimental design

NA

Validity of the findings

NA

Additional comments

NA

---

## Round 0.3 · accepted · Accept

In this revised version of the manuscript, the authors addressed the reviewer's comments. bHLH TFs are known to play a diverse physiological role in plants. In this MS, the authors identify the bHLH TFs in R. persica, and also provide substantial data suggesting the interaction of bHLH, CBF, and jasmonate pathway.